# Estimation of emissions from biomass burning in China (2003–2017) based on MODIS fire radiative energy data

Lifei Yin[1], Pin Du[1], Minsi Zhang[2], Mingxu Liu[1], Tingting Xu[1], Yu Song[1]

[1]State Key Joint Laboratory of Environmental Simulation and Pollution Control, Department of Environmental Science,
Peking University, Beijing, China

[2]National Center for Climate Change Strategy and International Cooperation (NCSC), Beijing, China

*Correspondence to:* Yu Song (songyu@pku.edu.cn)

**Abstract.** Biomass burning plays a significant role in air pollution and climate change. In this study, we used the method based on fire radiative energy (FRE) to develop a biomass burning emission inventory for China from 2003 to 2017. Daily fire radiative power (FRP) data derived from 1 km MODIS Thermal Anomalies/Fire products (MOD14/MYD14) were used to calculate FRE and combusted biomass. Available emission factors were assigned to four biomass burning types: forest, cropland, grassland and shrubland fires. The farming system and crop types in different temperate zones were taken into account in this research. Compared with traditional methods, the FRE method was found to provide a more reasonable estimates of emissions from small fires. The estimated average annual emission ranges, with a 90 % confidence interval, were 91.4 (72.7–108.8) Tg $CO_2$ yr$^{-1}$, 5.0 (2.3–7.8) Tg CO yr$^{-1}$, 0.24 (0.05–0.48) Tg $CH_4$ yr$^{-1}$, 1.43 (0.53–2.35) Tg NMHC yr$^{-1}$, 0.23 (0.05–0.45) Tg $NO_x$ yr$^{-1}$, 0.09 (0.02–0.17) Tg $NH_3$ yr$^{-1}$, 0.03 (0.01–0.05) Tg $SO_2$ yr$^{-1}$, 0.04 (0.01–0.08) Tg BC yr$^{-1}$, 0.27 (0.07–0.49) Tg OC yr$^{-1}$, 0.51 (0.19–0.84) Tg $PM_{2.5}$ yr$^{-1}$, 0.57 (0.15–1.05) Tg $PM_{10}$ yr$^{-1}$. Forest fires are determined to be the primary contributor to open fire emissions, accounting for 45 % of the total $CO_2$ emissions (average 40.8 Tg yr$^{-1}$). Crop residue burning ranked for the second places with a large portion of 39 % (average 35.3 Tg yr$^{-1}$). During the study period, emissions from forest and grassland fires showed a significant downward trend. Crop residue emissions continued to rise during 2003–2015 but dropped by 42 % in 2015–2016. Emissions from shrubland were negligible and little changed. Forest and grassland fires are concentrated in northeastern China and southern China, especially in dry season (from October to March of the following year). Plain areas with high crop yields, such as the North China Plain, experienced high agricultural fire emissions in harvest seasons. Most shrubland fires located in Yunnan and Guangdong province. The resolution of our inventory (daily, 1

km) is much higher than previous inventories, such as GFED4s and GFASv1.0. It could be used in global and regional air quality modeling.

## 1. Introduction

Biomass burning is an important source of gaseous and particulate matter emissions to the troposphere (Crutzen et al., 1979;Seiler and Crutzen, 1980). Globally, biomass burning contributes around 20 %–30 % of $CO_2$ emissions and chemically active gases such as hydrocarbons, CO and $NO_x$ (Andreae, 1991), approximately 42 % of black carbon (BC), and 74 % of primary organic carbon (OC) (Bond et al., 2004). These compounds have significant impacts on air quality, atmospheric chemistry, climate change, and human health (Andreae et al., 1994;Reid et al., 2005).

In China, the annual amount of crop residue burned in fields estimated by Streets et al. (2003) was 110 Tg, accounting for 44 % of all crop residue burned in Asia, leading to substantial pollutant emissions. Emission from other types of biomass burning, such as forest fires, are also of great concern (Chen et al., 2017a). Early studies used provincial statistical data to estimate biomass burning emissions. This method required many parameters that depend on local environment or agricultural practices and could vary greatly in different research, leading to significant emission uncertainties (Liu et al., 2015). Studies statistically evaluated fire emissions in China with results of annual $CO_2$ emissions of 68–150 Tg from crop residue burning (Ni et al., 2015;Huang et al., 2012;Li et al., 2016b) and 3–40 Tg from forest fires (Lu et al., 2006;Yan et al., 2006). This approach produced emission estimates at a coarse resolution that cannot be used for detailed analysis of spatiotemporal patterns. Thus, two methods based on remote sensing data has been increasingly used. The first one is based on fire count data provided by active fire products. In this approach, a maximum burned area of 1 $km^2$ is assumed for each fire count detected. Mehmood et al. (2018) calculated the mean emission of $CO_2$ for the period of 2002–2016 as 160 Tg $yr^{-1}$ (with 24 Tg from crop residue burning) by using data derived from the Fire INventory from NCAR version 1.5 (FINNv1.5), which was established by the fire count method (Wiedinmyer et al., 2011). Because the actual area burned of each fire count could vary to a large extent, using fire counts as a proxy for fire-affected area may lead to great potential error on emission estimates (Song et al., 2009). The other one is based on the burned areas products (Song et al., 2010). The estimated emission is a product of burned area ($km^2$), aboveground biomass density burned in fields (kg dry matter $m^{-2}$), combustion efficiency (%), and emission factor (g

kg$^{-1}$) for each pollutant. Generally, the uncertainty originates from all of the above factors. Moreover, as the average cultivated area of a farming household is very limited in China (around $10^4 \, m^2$), each agricultural fire burns within a small extent (Liu et al., 2015). Therefore, the fire count method is likely to overestimate the burned area of crop residue burning, and these fires are not detected efficiently by the available burned area algorithms due to the small areas and intermittency (Song et al., 2009).

For a better estimation of biomass burning emission, an approach based on fire radiative energy (FRE) was proposed as a new tool for global studies of vegetation fires around the year 2000 (Kaufman et al., 1996;Wooster, 2002). FRE is the amount of energy radiated during the combustion process (Kaufman et al., 1996). The fuel mass consumed could be calculated by multiplying FRE by a conversion ratio, which has been demonstrated to be insensitive to vegetation type and could be treated as a constant (Freeborn et al., 2008;Wooster, 2002). The FRE method estimates biomass consumed according to energy

radiated from fires, which could avoid the uncertainty caused by inaccuracy of satellite-derived burned area and therefore improve the estimation, especially for small fire emissions. Moreover, the amount of pollutants released by biomass burning could be calculated as a product of FRE, conversion ratio and emission factors, reducing uncertainties from multiple parameters that are not reliably defined at regional and global scales (Wooster et al., 2005). Liu et al. (2015) applied FRE approach to estimate emissions of crop residue burning in North China Plain during the harvest season (June). The differences of their

results with those based on official statistical data (Huang et al., 2012) were mostly around −13 % with the largest difference of −49 %. Besides, their results were significantly higher than those derived from burned area product (MCD45A1). These comparisons suggested that the approach produced a reasonable estimation.

According to the accumulated temperature, China is divided into six temperature zones (tropical zone, subtropical zone, warm-/middle-/cold–temperate zone and Qinghai-Tibet plateau) (Shi, 2015). The growth period and main crop type varies among

temperature zones. For example, in tropical regions, the main crops are rice, sugarcane and natural rubber, and rice grown there could be harvested for three times per year. While in middle-temperate zone, the main crops are spring wheat, maize and soybean, which ripen only once a year. Liu et al. (2015) focused on emissions from winter wheat residue burning in June, the result of which is not suitable for the whole country. Some studies have used FRE method to estimate global and regional biomass burning emissions (Vadrevu et al., 2011;McCarty et al., 2012;Vermote et al., 2009). However, to our knowledge, few

studies in China used this approach to estimate emissions from crop residue burning and other vegetation fires on a national

scale. Thus, the establishment of a biomass burning emission inventory based on FRE method for the whole country is of great significance.

In this study, we used FRP data derived from the MODIS active fire products to calculate emissions of 11 pollutants from biomass burning in China (excluding fires occurring on the small islands in the South China Sea) for the period of 2003–2017.

The spatiotemporal distribution of emissions from four biomass burning types (forest, grassland, cropland, and shrubland fires) were detailed studied. A daily gridded 1 km emission inventory of biomass burning was established; this inventory could meet the requirements of global and regional air quality simulations.

## 2. Methods and data

### 2.1 Methods

Pollutant emissions were calculated as the product of dry mass burned (kg) and a corresponding emission factor (g kg$^{-1}$). In this study, emission factors for each land cover type were obtained from previous publications (Table S1). If more than one value for an emission factor is available, the average value is used.

The amount of biomass consumed was calculated by multiplying FRE by a conversion ratio, which was not significantly influenced by vegetation types (Wooster et al., 2005):

$M = FRE \times CR$ (1)

Where $M$ is the dry biomass consumed of one grid cell, $FRE$ is the total radiative energy during the fire lifespan for one grid cell, and $CR$ is the conversion ratio (kg MJ$^{-1}$) used to convert FRE to combusted biomass.

Wooster et al. (2005) reported a conversion ratio of $0.368\pm0.015$ kg MJ$^{-1}$, and that evaluated by Freeborn et al. (2008) was $0.453\pm0.068$ kg MJ$^{-1}$. In this study, we used the average value (0.411 kg MJ$^{-1}$).

FRE was estimated by integrating FRP (i.e. instantaneous FRE) over the duration of the fire process. In this study, FRP data from MODIS active fire products (MOD14/MYD14) were used. The MODIS sensors, onboard the polar-orbiting satellites Terra and Aqua, acquire four discrete FRP data at 1030/2230 (Terra) and 0130/1330 (Aqua), equatorial local time. Therefore, the fire diurnal variation cannot be directly detected by satellite observation and many fire events have been missed. To

calculate FRE and make up the omission error, we used a modified Gaussian function (Vermote et al., 2009) to parameterize the FRP diurnal cycle. This parameterization describes the discrete observations as a continuous function and simplifies integral process to calculate total fire energy released. The modified Gaussian function is:

$$FRE = \int FRP = \int_0^{24} FRP_{peak} (b + e^{-\frac{(t-h)^2}{2\sigma^2}}) \, dt \tag{2}$$

Where $FRP_{peak}$ represents the peak of the diurnal cycle, $b$ represents the background FRP, $\sigma$ represents the standard deviation of the curve, and $h$ represents the hour of peak FRP.

Monthly mean Terra and Aqua FRP (T/A ratio) was used to determine the required parameters with following equations (Vermote et al., 2009) :

$$b = 0.86x^2 - 0.52x + 0.08 \tag{3}$$

$$\sigma = 3.89x + 1.03 \tag{4}$$

$$h = -1.23x + 14.57 \tag{5}$$

$$FRP_{peak} = \frac{FRP_{Aqua\ Day}}{[b + e^{-\frac{(13.5-h)^2}{2\sigma^2}}]} \tag{6}$$

Where $x$ represents the T/A ratio. We found that the original parameterized FRP diurnal cycle could not agree well with the observed FRP temporal variation in China, possibly due to inaccurate FRP peak hour. Because it has been pointed that $h$ has little effect on the final result of FRE (Vermote et al., 2009), we added a parameter $\varepsilon$ ($\varepsilon=4$) in order to modify FRP peak hour (Liu et al., 2015). The modified equation was:

$$h = -1.23x + 14.57 + \varepsilon \tag{7}$$

Monthly mean T/A ratio were calculated for each type of biomass burning. Different combustion characteristics of fuel types could be reflected by specific T/A ratio. As shown in Fig.S1 (excluding small islands in the South China Sea), China is divided into six temperature zones (tropical zone, subtropical zone, warm-/middle-/cold–temperate zone and Qinghai-Tibet plateau). Because the dominate crop types vary greatly among temperature zones, we calculated T/A ratio for each zone separately. Using respective T/A ratio to calculate factors required in Eq. (2), the FRP diurnal cycle was parameterized for each zone and harvest season, which could reflect specific combustion characteristic of different straw types.

## 2.2 Data

The MODIS Thermal Anomalies/Fire 5-Min L2 Swath Products (MOD14/MYD14) are primarily derived from MODIS 4- and 11-micrometer radiances. The products provide the fire occurrence, location, FRP and other information of fire events with moderate spatial resolution (1 km$^2$) and high temporal resolution (daily). MOD14 data were obtained from Terra, which passes at 10:30 and 22:30 local time (LT), and MYD14 data were provided by Aqua, which acquires observations at 01:30 and 13:30 (LT). If Terra and Aqua detected the same fire events (determined by the time and location of fire occurrence), we would use information from Aqua since there is almost no difference between Terra and Aqua data and choosing Aqua can support the $FRP_{peak}$ calculation. We used data for a 15-year period (2003–2017) to calculate FRE and estimate emissions.

The GlobeLand30 dataset maps global land cover at 30 m spatial resolution in two base years (2000 and 2010) (Chen et al., 2017b), as shown in Fig S2 (small islands in the South China Sea are not included). GlobeLand30 data are generated by multispectral images derived from Landsat TM, ETM+ and Chinese Environmental Disaster Alleviation Satellite (HJ-1). The result of accuracy assessment shows that the overall accuracy of GlobeLand30 reaches 83.5 %. GlobeLand30 dataset consists of 10 land cover types, namely cultivated land, forest, grassland, shrubland, wetland, water bodies, tundra, artificial surfaces, bareland, permanent snow and ice. In this study, the land cover types are characterized by GlobeLand30-2000 for years 2003–2005 and GlobeLand30-2010 for years 2006–2017. We combined the land-cover map of China and the latitude and longitude data of fire count in MOD14/MYD14 to determine the biomass fuel types. For instance, if a fire count locates in cropland area, it will be considered as a crop residue burning event.

To compare the results, we computed open fire emissions using data derived from MODIS burned area products (MCD64A1, http://modis-fire.umd.edu/), the fourth version of the Global Fire Emission Database (with small fires) (GFED4s), Global Fire Assimilation System (GFASv1.0), and FINNv1.5 (http://bai.acom.ucar.edu/Data/fire/). We derived data for 2003–2017 from MCD64A1, which is a monthly, global gridded 500 m product containing per-pixel burned area information. GFED4s provides monthly emission data at a spatial resolution of 0.25°; the latest GFED4s data are for 2016. GFASv1.0 calculates daily biomass burning emissions by assimilating FRP data from MODIS sensors on a global 0.5°×0.5° grid; we used GFASv1.0 data to estimate emission from 2003 to 2013. FINNv1.5 provides daily high-resolution (1 km) emissions of global biomass burning; data from 2003 to 2016 were used for comparison in this study.

## 3. Results and discussion

A total of 462,525 biomass fire pixels were detected by Terra, and 492,822 by Aqua from 2003 to 2017. When a fire pixel was probed by both satellites within the same day, the Terra pixel was removed to avoid repeated computations. Thus, a total of 942,933 fire pixels were applied to estimate emissions. The inter-annual variation in emissions was shown in Table 1. For the
15-year study period, average emissions of $CO_2$, $CO$, $CH_4$, NMHC, $NO_x$, $NH_3$, $SO_2$, BC, OC, $PM_{2.5}$, and $PM_{10}$ were estimated to be 91.4, 4.0, 0.24, 1.43, 0.23, 0.09, 0.03, 0.04, 0.27, 0.51 and 0.57 Tg $yr^{-1}$, respectively. Taking $CO_2$ emission as an example, the maximum emission occurred in 2003 (123.0 Tg), followed by 2014 (117.3 Tg), and the minimum emission occurred in 2016 (59.8 Tg). These results will be discussed in detail in Section 3.2.

### 3.1 Spatial distribution of emissions

Average annual emissions of 11 pollutants at the provincial level were listed in Table 2, and source-specific emissions of $CO_2$ for each province were presented in Fig.1. Using $CO_2$ as a representative example, southwestern China and northeastern China contribute most to the total emission, with portion of 28 % and 26 %, respectively. On a national scale, forest fires contribute the largest portion (45 %) of total $CO_2$ emissions from open fires. Agricultural fires and grassland fires ranked for the second and third places, accounting for 39 % and 15 %, respectively. Regionally, the main emission contributor is different. In
southwestern region, the percentage of emission from forest fires could reach up to 65 %, whereas the most important source in northeastern China is crop residue burning, accounting for 47 % of total emissions. The result was in connection with rural population intensity and land use patterns (Qiu et al., 2016). For example, due to the dense boreal forests and developed agriculture, the highest emission was found in Heilongjiang with 46 % from agriculture fires and 54 % from forest and grassland fires. Similarly, in the southwestern region, the dense vegetative cover of Yunnan-Guizhou Plateau greatly
contributes to fire events. Benefiting from fertile land and favorable climate, northern and central regions contain many principal agricultural provinces (including Shandong, Henan, Hubei and Anhui Provinces) and therefore large amounts of crop residue were burned in field during the harvest season, contributing 55 % to the total emissions. Southeastern provinces in the Middle-Lower Yangtze River Plain and the southeastern hills have abundant cultivated land and forest resources, resulting in relatively high $CO_2$ emissions from cropland and forest fires (with portion of 32 % and 56 %, respectively). Northwestern

China experience extremely dry weather, which leads to low vegetative cover and negligible emissions from biomass burning. For instance, annual mean $CO_2$ released from open fires in Ningxia and Qinghai were 0.21 Tg and 0.13 Tg, respectively. Vegetation in these areas mainly consists of grass and a few drought-resistant crops; hence, an extremely high proportion (92 %) of $CO_2$ emissions in the northwest arose from grassland and cropland fires.

Nationwide spatial patterns of $CO_2$ emissions from four sources were shown in Fig. 2 (biomass fire emissions from the small islands in the South China Sea are not included). Forest and grassland fire emissions were mainly distributed in northeastern China and southern China. Dense vegetative covers in Yunnan-Guizhou Plateau, Inner Mongolian Plateau, Daxing'anling, Xiaoxing'anling and the southeast hills greatly contribute to fire events. Cropland fire emissions were concentrated in the three great plains of China, namely the Northeast China Plain, the North China Plain, and the Middle–Lower Yangtze Plain. Because of high crop production in these areas, large quantities of agricultural residues were burned in fields during the short period following the harvest season. In addition, due to snowmelt in the Tianshan Mountains, there are many oases located at the foot of the mountain range in Xinjiang Province. These oases are suitable for growing crops such as wheat and maize (Zhou et al., 2017). Therefore, crop fire emissions in Xinjiang province were higher than those in other northwestern provinces. Compared with other fire types, emissions from shrubland fire were negligible and the high emissions were concentrated in Guangdong and Yunnan province.

**3.2 Temporal pattern of emissions**

The annul variations of total and source–specific $CO_2$ emissions were presented in Fig.3. Peak emissions occurred in 2003, 2009 and 2014; forest fires in 2003 and 2009, and cropland fires in 2014 were determined to be the primary contributors, accounting for 61 %, 56 %, and 49 % of total emissions in that year, respectively. Our results were in accordance with the records reported in official statistics. According to the China Forestry Statistical Yearbook, there are seven extraordinarily serious fire accidents in 2003, resulting in the largest forest burned area during the study period. A total of 35 serious fire accidents happened in 2009, 171 % higher than the 15-year average number of that kind of fire events (12.9). As over 95 % of forest fires in China are caused by human activities, the implement of strict forest conservation policies and the development of fire control technology contribute significantly to the emission decline (Huang et al., 2011). Forest fires are well controlled

after 2003 and emissions decreased by 78 % during the study period (from 74.7 Tg in 2003 to 16.6 Tg in 2017). Pollutants released by crop straw burning continue to rise in 2003–2014, leading to a peak emission of 57.6 Tg $CO_2$ in 2014. Because crop residues burning in field could be well controlled by strict supervision, cropland emissions have decreased rapidly in 2015–2016 (dropped by 42 %). However, the emissions increased again by 37 % in 2017. This variation trend was similar to that concluded by studies based on statistical data (Li et al., 2016b;Jian et al., 2018). Yan et al. (2006) pointed that as the socioeconomic development, which results in a decline of biofuel (crop residue, fuel wood) demand, crop residue is increasingly being burned in the field. Tao et al. (2018) found that the consumption of crop residues as residential energy in rural China decreased by 51 % from 1992 to 2012. We noted that the number of agricultural fire count increased by a factor of 3 in 2003–2014 (from 13683 to 67143), which could support the conclusion as well. Although the controlling of pollutants from crop residue burning in China started from 1965, it seems to be ineffective and the crop straw burning should be further focused. Emissions from grassland fires dropped by 60 % from 2003 to 2017 due to the conservation and supervision measures. Shrubland fire emissions were much lower than other fire emissions (range from 0.5 to 2.3 Tg $yr^{-1}$) and remained relatively stable during the study period. Emissions from forest, grassland and shrubland exhibited a small peak in 2014. According to the statistics, the total burned areas in 2014 for both forest and grassland are higher than previous years. The rise in burned area and emission could be attributed to an unusual warm condition occurred in 2014, which could facilitate the occurrence and spread of fires (Bond et al., 2015).

Seasonal variations of $CO_2$ emissions from each source were presented in Fig.4. In terms of total emissions, spring (March, April and May) contributed the most emissions due to the impact of dry weather. The lowest emissions occurred in rainy season including July, August, and September, producing 2.1, 1.7, and 1.8 Tg $CO_2$, respectively. From the perspective of source-specific emissions, forest and grassland fires exhibited similar temporal variation, i.e., higher emissions in winter and spring, and lower emissions in summer. The highest emissions from forest and grassland fires occurred in the period of January to May. This pattern was strongly affected by favorable fire conditions such as low vegetation moisture content and high wind speed (Song et al., 2009). In addition, Li et al. (2015) found that a large portion of forest fires in spring were induced by sacrificial activity in Tomb-sweeping Day (April 5). Forest Fires in winter were concentrated in southern China due to the impacts of low precipitation and mild temperatures. In contrast, boreal forests rarely burned because of the low temperatures

and moist snow cover. This result was consistent with the that reported by Chen et al. (2017a). The temporal distribution of shrubland fire emissions was also similar to that of forest and grassland fires, but emissions from bush only account for a small fraction of total levels (approximately 1 %). Emissions from crop burning were closely related to agriculture activities. Different main crops and sowing/harvest times in different areas lead to multiple emission peaks (Jin et al., 2018). Highest

emissions occurred in summer, and small peaks were detected in spring and autumn. Emissions from agriculture fires contribute 84 % to total emissions in summer, which were concentrated in June due to the large amount of winter wheat straw burning in the North China Plain. From March to May, as large amounts of crop residues were burned to clear the cultivated land for sowing, fires were scattered throughout the country. In autumn (especially October), corn straws burning in the Northeast China Plain and late rice residue burning in southern China were primary contributors, and small areas of maize

residue burning could be found in northern China (Chen et al., 2017a). During winter, crop burning mostly occurred in southern China due to citrus harvest and orchard clearing activity.

### 3.3 Comparison with other studies

The average annual emission estimates calculated in this study were compared to those based on data from the burned area product (MCD64A1), GFED4s, GFASv1.0, and FINNv1.5 (Table 3). Generally, our results were closed to those derived from

GFED4s and GFASv1. However, as shown in Table 3, results calculated by using data from burned area product MCD64A1 were substantially underestimated. In this method, burned area is one of the most important factors in calculating emissions, so that the underestimation could be attributed to omission of fires with small areas and short duration (Song et al., 2009). Emission estimates by FINNv1.5 were higher than those of this study with a difference ranging from 29 % to 194 %.

The comparison of annual mean $CO_2$ emission from each fire type in our study with those derived from other methods was

listed in Table 4 (shrubland and grassland fires are lumped into one category in GFED4s). When compared with results of Huang et al. (2012) and Yan et al. (2006), which were based on official statistical data, our results were larger for forest and grassland fires, and underestimated for crop residue burning. According to Yan et al. (2006), forest and grassland fires were understated in statistics for both personal and political reasons. They suggested that satellite data are preferable to statistical data to estimate emissions from forest and grassland fires. When statistics were used to estimate crop residue emission, the

amount of crop residue consumed are calculated as a product of crop production, residue-to-production ratio, dry matter-to-crop residue ratio, the percentage of dry matter burned in fields, and combustion efficiency. Values of these parameters depend on local agricultural practices and vary greatly in different studies. For example, the value of percentage of residue burned in field, which is one of the most important factors to be determined, ranges from 6.6 % to 82 % in different research (Gao et al., 2002;Yang et al., 2008;Yan et al., 2006). The accumulation of uncertainties derived from multiple factors could result in significant emission uncertainties. Using statistical data, amount of burned residue was estimated to be 40–160 Tg yr$^{-1}$, showing a great potential error (Li et al., 2016a;Huang et al., 2012). Therefore, results derived from statistical are not necessarily reliable. When compared to other inventories based on remote sensing data, our results agreed well with those reported by GFED4s and were substantially higher than those derived from burned area product (MCD64A1). Datasets in GFED4s are based on burned area boosted by small fire burned area, which could provide a relatively high emission estimation of agricultural fires. Due to shielding by the dense canopy (Moreira de Araújo et al., 2012;Roy and Boschetti, 2009) and higher small-fire omission rates, emissions derived from burned area product (MCD64A1) were underestimated by 33 %–93 %, especially for forest fire (−85 %) and cropland fire (−93 %) emissions. FINNv1.5 emission estimates were higher for forest and shrubland fires. The discrepancy can primarily be attributed to the overestimation of burned area of forest fires (Roy et al., 2008) and different land cover characterization maps used. Estimates of grassland and cropland fire emissions in FINNv1.5 were closed to our results, with differences of 3 % and 8 %, respectively.

In conclusion, our estimates were higher than those based on statistics for forest and grassland fire emissions, but lower for crop residue burning emission. Our results were higher than those based on burned area products as the FRE method avoids uncertainties cause by inaccuracy of satellite-derived burned area and multiple other parameters. The results were closed to those derived from FINNv1.5 in terms of emissions from grassland and cropland fires and accorded with those from GFED4s for all fire types. The temporal and spatial resolution of our inventory (daily, 1 km) are higher than that of GFED4s (monthly, 0.25 degrees) and GFASv1.0 (daily, 0.5 degrees). Compared with other inventories, we considered specific combustion characteristics of different crop types and calculated the agriculture fires emissions separately according to the distribution of temperate zones. Therefore, this method developed a high-resolution inventory and improved estimation of biomass burning emissions, especially for small fires in cropland.

## 4 Uncertainty

Several sources of error impact the accuracy of our estimate. The first error source is related to the radiative energy diurnal cycle parameterization that impacts the calculation of FRE. In addition, the error in the fire detection and empirical formula for computing FRP have a considerable impact on the accuracy of FRE. The use of the conversion ratio in order to convert FRE to combusted biomass is one of error sources as well. Since emission factors vary in time and space, they could also bring large uncertainties. In this study, we considered errors of three independent variables, namely FRE, conversion ratio and emission factors. According to the error budget suggested by Vermote et al. (2009), we assumed that the relative error of FRE and the conversion ratio was 31 % and 10 %, respectively. The uncertainty of the emission factor is species dependent and we applied the uncertainty suggested in Huang et al. (2012) , as shown in Table S2. We ran 20,000 Monte Carlo simulations to estimate the range of average annual fire emissions in 2003–2017 with a 90 % confidence interval. In Monte Carlo simulation, random number were selected from normal distribution of input variables. Estimated emissions of $CO_2$, CO, $CH_4$, NMHC, $NO_x$, $NH_3$, $SO_2$, BC, OC, $PM_{2.5}$ and $PM_{10}$ were 91.4 (72.7–108.8), 5.0 (2.3–7.8), 0.24 (0.05–0.48), 1.43 (0.53–2.35), 0.23 (0.05–0.45), 0.09 (0.05–0.17), 0.03 (0.01–0.05), 0.04 (0.01–0.08), 0.27 (0.07–0.49), 0.51 (0.19–0.84), and 0.57 (0.15–1.05) Tg $yr^{-1}$, respectively.

## 5 Conclusion

In this study, we developed a high-spatiotemporal-resolution (daily data in a 1 km×1 km grid) inventory of emissions from biomass burning in China based on MODIS FRP data. The annual average emissions of were 91.4 (72.7–108.8), 5.0 (2.3–7.8), 0.24 (0.05–0.48), 0.23 (0.05–0.45), 0.04 (0.01–0.08), 0.27 (0.07–0.49) and 0.51 (0.19–0.84) Tg $yr^{-1}$ for $CO_2$, CO, $CH_4$, $NO_x$, BC, OC, and $PM_{2.5}$, respectively. On a national scale, forest fires contribute the largest portion (45 %) of total $CO_2$ emissions from open fires. Agricultural fires and grassland fires ranked for the second and third places, accounting for 39 % and 15 %, respectively. Emissions in southwestern China and northeastern China are determined to be primary contributor, accounting for 52 % of the total emission. Spatially, forest and grassland fires were concentrated in the northeast and south regions. Cropland fires extensively occurred in the Northeast China Plain, the North China Plain, and the Middle–Lower Yangtze Plain,

and shrubland fires happened in the south region such as Guangdong and Yunnan province. Temporally, total emissions were relatively high in 2003 and 2014, and the lowest emissions occurred in 2016. Most wild fires, including forest, grassland and shrubland, occurred during dry season (October to March of the following year), whereas agricultural fires were concentrated in the harvest season (June and October). Compared with estimations by other methods, our results are much higher than those obtained from the burned area method as the FRE method avoids uncertainties cause by inaccuracy of satellite-derived burned area and multiple other parameters. Our estimates were very close to those from GFED4s and GFASv1.0, as well as grassland and cropland fire emissions from FINNv1.5, indicating that our results were reasonable and can be used for further research. Furthermore, the temporal and spatial resolution of our inventory (daily, 1 km) are higher than that of GFED4s (monthly, 0.25 degrees) and GFASV1.0 (daily, 0.5 degrees). Uncertainties in our estimates may have been caused by many factors such as the characterization of the fire energy radiative diurnal cycle; thus, future studies should seek to improve the accuracy of the method.

*Data availability.* MODIS data can be freely accessed at https://search.earthdata.nasa.gov/search. GlobeLand30 data are downloaded from http://www.globallandcover.com/GLC30Download/index.aspx. GFASv1.0 data are available on http://apps.ecmwf.int/datasets/data/cams-gfas/. GFED4s data can be downloaded from https://daac.ornl.gov/VEGETATION/guides/fire_emissions_v4.html. FINNv1.5 data can be found at http://bai.acom.ucar.edu/Data/fire/.

*Author contributions.* This work was designed by YS and performed by LY, PD, MZ, ML, and TX. LY and YS led the writing of the papers and prepared the figures with contributions from all co-authors.

*Competing interests.* The authors declare that they have no conflict of interest.

*Acknowledgements.* The MODIS Thermal Anomalies/Fire products (MOD14/MYD14) and burned area product (MCD64A1) were provided by Land Process Distributed Active Archive Center (LPDAAC), USA. The GlobeLand30 land cover dataset

was provided by the National Geomatics Center of China. This study was funded by National Key R&D Program of China (2016YFC0201505) and National Natural Science Foundation of China (NSFC) (91644212 and 41675142).

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

**Table 1. Biomass burning emissions inventory (Tg) of China from 2003 to 2017.**

| Year | $CO_2$ | CO | $CH_4$ | NMHC | $NO_x$ | $NH_3$ | $SO_2$ | BC | OC | $PM_{2.5}$ | $PM_{10}$ |
|------|------|-----|------|------|------|------|------|------|------|------|------|
| 2003 | 123.0 | 6.4 | 0.27 | 1.18 | 0.27 | 0.09 | 0.04 | 0.05 | 0.42 | 0.75 | 0.84 |
| 2004 | 113.0 | 6.0 | 0.27 | 1.44 | 0.27 | 0.10 | 0.04 | 0.05 | 0.36 | 0.66 | 0.74 |
| 2005 | 74.6 | 4.1 | 0.19 | 1.12 | 0.18 | 0.07 | 0.02 | 0.03 | 0.22 | 0.42 | 0.47 |
| 2006 | 91.6 | 4.9 | 0.22 | 1.22 | 0.22 | 0.08 | 0.03 | 0.04 | 0.29 | 0.53 | 0.59 |
| 2007 | 84.2 | 4.6 | 0.22 | 1.30 | 0.21 | 0.08 | 0.03 | 0.04 | 0.26 | 0.48 | 0.53 |
| 2008 | 97.6 | 5.2 | 0.23 | 1.22 | 0.23 | 0.08 | 0.03 | 0.04 | 0.32 | 0.59 | 0.64 |
| 2009 | 101.3 | 5.4 | 0.24 | 1.30 | 0.24 | 0.09 | 0.03 | 0.04 | 0.33 | 0.60 | 0.66 |
| 2010 | 87.4 | 4.7 | 0.22 | 1.24 | 0.21 | 0.08 | 0.03 | 0.04 | 0.27 | 0.50 | 0.55 |
| 2011 | 77.2 | 4.3 | 0.20 | 1.27 | 0.20 | 0.07 | 0.02 | 0.03 | 0.22 | 0.42 | 0.47 |
| 2012 | 81.1 | 4.6 | 0.23 | 1.57 | 0.22 | 0.08 | 0.02 | 0.04 | 0.21 | 0.41 | 0.47 |
| 2013 | 93.2 | 5.2 | 0.24 | 1.49 | 0.24 | 0.09 | 0.03 | 0.04 | 0.26 | 0.50 | 0.57 |
| 2014 | 117.3 | 6.6 | 0.33 | 2.16 | 0.31 | 0.12 | 0.03 | 0.05 | 0.31 | 0.61 | 0.69 |
| 2015 | 94.8 | 5.5 | 0.28 | 2.03 | 0.27 | 0.10 | 0.03 | 0.04 | 0.22 | 0.46 | 0.53 |
| 2016 | 59.8 | 3.4 | 0.17 | 1.20 | 0.17 | 0.06 | 0.02 | 0.03 | 0.14 | 0.30 | 0.34 |
| 2017 | 75.1 | 4.4 | 0.23 | 1.66 | 0.22 | 0.08 | 0.02 | 0.03 | 0.17 | 0.36 | 0.41 |
| Average | 91.4 | 5.0 | 0.24 | 1.43 | 0.23 | 0.09 | 0.03 | 0.04 | 0.27 | 0.51 | 0.57 |

**Table 2. Average biomass burning emissions (Gg) in each province from 2003 to 2017.**

| region/province | $CO_2$ | CO | $CH_4$ | NMHC | $NO_x$ | $NH_3$ | $SO_2$ | BC | OC | $PM_{2.5}$ | $PM_{10}$ |
|---|---|---|---|---|---|---|---|---|---|---|---|
| **Northwest** | 2607.4 | 154.9 | 8.1 | 60.8 | 7.8 | 2.9 | 0.7 | 1.2 | 4.8 | 11.0 | 13.4 |
| Xinjiang | 1207.2 | 72.6 | 3.9 | 29.7 | 3.7 | 1.4 | 0.3 | 0.5 | 2.0 | 4.8 | 6.0 |
| Gansu | 359.0 | 21.1 | 1.1 | 8.0 | 1.1 | 0.4 | 0.1 | 0.2 | 0.7 | 1.6 | 1.9 |
| Ningxia | 211.7 | 13.0 | 0.7 | 5.6 | 0.7 | 0.3 | 0.1 | 0.1 | 0.3 | 0.8 | 1.0 |
| Qinghai | 131.4 | 7.5 | 0.4 | 2.5 | 0.4 | 0.1 | 0.0 | 0.1 | 0.2 | 0.5 | 0.7 |
| Shaanxi | 698.0 | 40.8 | 2.1 | 15.0 | 2.0 | 0.8 | 0.2 | 0.3 | 1.5 | 3.3 | 3.8 |
| **Northeast** | 23524.0 | 1323.4 | 64.8 | 416.9 | 63.6 | 22.8 | 6.9 | 10.2 | 56.9 | 114.6 | 136.0 |
| Inner Mongolia | 5769.3 | 307.0 | 14.1 | 70.5 | 14.5 | 4.6 | 1.8 | 2.3 | 15.0 | 28.5 | 35.3 |
| Heilongjiang | 13812.4 | 775.1 | 37.7 | 241.7 | 36.8 | 13.4 | 4.1 | 6.1 | 34.9 | 69.5 | 81.1 |
| Jilin | 2394.4 | 148.4 | 8.1 | 67.0 | 7.6 | 3.0 | 0.6 | 1.1 | 3.9 | 9.8 | 11.6 |
| Liaoning | 1548.0 | 92.9 | 4.9 | 37.8 | 4.6 | 1.8 | 0.4 | 0.7 | 3.0 | 6.9 | 8.1 |
| **North** | 8336.8 | 516.3 | 28.2 | 232.4 | 26.2 | 10.5 | 2.2 | 4.0 | 14.8 | 35.8 | 41.2 |
| Beijing | 146.2 | 8.8 | 0.5 | 3.7 | 0.4 | 0.2 | 0.0 | 0.1 | 0.3 | 0.7 | 0.8 |
| Shanxi | 1153.7 | 66.6 | 3.4 | 23.5 | 3.3 | 1.2 | 0.3 | 0.5 | 2.5 | 5.4 | 6.4 |
| Hebei | 1592.5 | 97.3 | 5.2 | 42.0 | 4.9 | 1.9 | 0.4 | 0.8 | 2.9 | 6.9 | 8.0 |
| Shandong | 2258.5 | 143.6 | 8.0 | 69.5 | 7.4 | 3.0 | 0.6 | 1.1 | 3.5 | 9.2 | 10.5 |
| Tianjin | 205.2 | 13.2 | 0.7 | 6.6 | 0.7 | 0.3 | 0.1 | 0.1 | 0.3 | 0.8 | 0.9 |
| Henan | 2980.8 | 186.9 | 10.3 | 87.1 | 9.5 | 3.9 | 0.8 | 1.5 | 5.2 | 12.8 | 14.5 |
| **Central** | 15299.8 | 844.0 | 39.7 | 243.6 | 37.9 | 14.6 | 4.7 | 7.0 | 47.1 | 88.4 | 96.7 |
| Hubei | 1832.6 | 102.9 | 5.0 | 32.4 | 4.7 | 1.8 | 0.6 | 0.9 | 5.3 | 10.3 | 11.3 |
| Anhui | 4227.0 | 262.6 | 14.3 | 119.5 | 13.2 | 5.4 | 1.1 | 2.1 | 7.9 | 18.8 | 21.1 |
| Hunan | 5240.9 | 271.7 | 11.6 | 52.5 | 11.4 | 4.2 | 1.7 | 2.3 | 19.0 | 33.4 | 36.3 |
| Jiangxi | 3999.2 | 206.8 | 8.8 | 39.3 | 8.6 | 3.2 | 1.3 | 1.8 | 14.8 | 25.9 | 28.0 |
| **Southwest** | 25603.5 | 1326.1 | 56.6 | 249.8 | 55.8 | 20.1 | 8.4 | 11.1 | 92.0 | 160.4 | 175.1 |
| Xizang | 1993.7 | 100.2 | 4.1 | 14.3 | 4.1 | 1.4 | 0.7 | 0.9 | 7.7 | 13.1 | 14.3 |
| Sichuan | 2531.7 | 134.9 | 6.1 | 30.1 | 6.1 | 2.0 | 0.8 | 1.0 | 7.6 | 13.4 | 15.5 |

| region/province | $CO_2$ | CO | $CH_4$ | NMHC | $NO_x$ | $NH_3$ | $SO_2$ | BC | OC | $PM_{2.5}$ | $PM_{10}$ |
|---|---|---|---|---|---|---|---|---|---|---|---|
| Chongqing | 261.0 | 15.8 | 0.8 | 6.7 | 0.8 | 0.3 | 0.1 | 0.1 | 0.5 | 1.2 | 1.4 |
| Yunnan | 10335.9 | 538.8 | 23.2 | 106.5 | 22.9 | 8.2 | 3.4 | 4.5 | 36.5 | 63.8 | 69.7 |
| Guizhou | 2460.2 | 126.1 | 5.4 | 21.8 | 5.5 | 1.8 | 0.8 | 1.0 | 8.2 | 14.5 | 16.6 |
| Guangxi | 8021.1 | 410.1 | 17.0 | 70.5 | 16.5 | 6.2 | 2.7 | 3.6 | 31.6 | 54.4 | 57.6 |
| **Southeast** | 16046.3 | 868.4 | 39.7 | 224.7 | 38.3 | 14.5 | 5.1 | 7.2 | 51.9 | 95.1 | 103.9 |
| Jiangsu | 2587.7 | 166.3 | 9.4 | 82.8 | 8.6 | 3.6 | 0.7 | 1.3 | 3.8 | 10.3 | 11.8 |
| Shanghai | 112.5 | 7.2 | 0.4 | 3.6 | 0.4 | 0.2 | 0.0 | 0.1 | 0.2 | 0.4 | 0.5 |
| Zhejiang | 1515.8 | 86.1 | 4.2 | 28.4 | 4.0 | 1.6 | 0.5 | 0.7 | 4.2 | 8.3 | 9.1 |
| Fujian | 3428.0 | 176.0 | 7.4 | 31.4 | 7.2 | 2.7 | 1.1 | 1.5 | 13.0 | 22.6 | 24.3 |
| Guangdong | 7659.7 | 392.9 | 16.5 | 68.5 | 16.3 | 5.9 | 2.5 | 3.3 | 28.2 | 48.9 | 53.3 |
| Macao | 0.5 | 0.0 | 0.0 | 0.0 | 0.0 | 0.0 | 0.0 | 0.0 | 0.0 | 0.0 | 0.0 |
| Hong Kong | 24.7 | 1.2 | 0.0 | 0.1 | 0.0 | 0.0 | 0.0 | 0.0 | 0.1 | 0.2 | 0.2 |
| Hainan | 515.6 | 27.1 | 1.2 | 5.8 | 1.1 | 0.4 | 0.2 | 0.2 | 1.9 | 3.4 | 3.6 |
| Taiwan | 201.7 | 11.5 | 0.6 | 3.9 | 0.6 | 0.2 | 0.1 | 0.1 | 0.5 | 1.0 | 1.1 |

**Table 3. Comparison of annual mean $CO_2$ emissions (Tg) from biomass burning calculated in our study with estimates made by other methods.**

| Year | This study | MCD64A1[a] | GFED4s[b] | GFASv1[c] | FINNv1.5[d] |
|------|-----------|-----------|-----------|-----------|-------------|
| 2003 | 123.0 | 21.4 | 112.9 | 138.6 | 161.2 |
| 2004 | 113.0 | 10.7 | 104.5 | 90.3 | 176.4 |
| 2005 | 74.6 | 9.5 | 71.9 | 67.0 | 157.1 |
| 2006 | 91.6 | 11.2 | 91.5 | 76.1 | 185.5 |
| 2007 | 84.2 | 11.4 | 90.0 | 78.3 | 196.2 |
| 2008 | 97.6 | 25.1 | 122.4 | 96.3 | 217.1 |
| 2009 | 101.3 | 15.1 | 100.3 | 77.8 | 256.3 |
| 2010 | 87.4 | 12.3 | 80.8 | 76.1 | 213.4 |
| 2011 | 77.2 | 9.4 | 94.8 | 63.3 | 188.0 |
| 2012 | 81.1 | 10.9 | 77.5 | 74.0 | 223.3 |
| 2013 | 93.2 | 9.6 | 74.9 | 61.5 | 221.9 |
| 2014 | 117.3 | 20.8 | 114.3 | | 157.4 |
| 2015 | 94.8 | 14.8 | 105.5 | | 122.2 |
| 2016 | 59.8 | 7.4 | 79.3 | | 175.7 |
| 2017 | 75.1 | 16.3 | | | |
| Average | 91.4 | 13.5 | 95.5 | 81.7 | 189.4 |

[a] Estimations based on MODIS burned area product (MCD64A1).

[b] GEFD4s estimated emissions based on burned area boosted by small fires burned area (Van Der Werf et al., 2017).

5   [c] GFASv1 calculated emissions with a global fire assimilation system based on FRP (Kaiser et al., 2012).

[d] FINNv1.5 was established by using fire count method (Wiedinmyer et al., 2011).

**Table 4. Comparison of annual average $CO_2$ emissions (Tg) from each fire type calculated in our study with estimates made by other methods.**

|  | Forest | Grassland | Shrubland | Cropland |
|---|---|---|---|---|
| This study | 40.8 | 14.1 | 1.2 | 35.3 |
| Huang et al. (2012)[a] |  |  |  | 68.0 |
| Yan et al. (2006)[a] | 3.4 | 0.3 |  | 185.0 |
| MCD64A1[b] | 6.0 | 4.4 | 0.8 | 2.5 |
| GFED4s[b] | 36.2 | 19.7 |  | 38.2 |
| FINNv1.5[b] | 105.4 | 14.5 | 31.4 | 38.1 |

[a] Emissions estimated by using statistical data.

[b] Refer to Table 3.

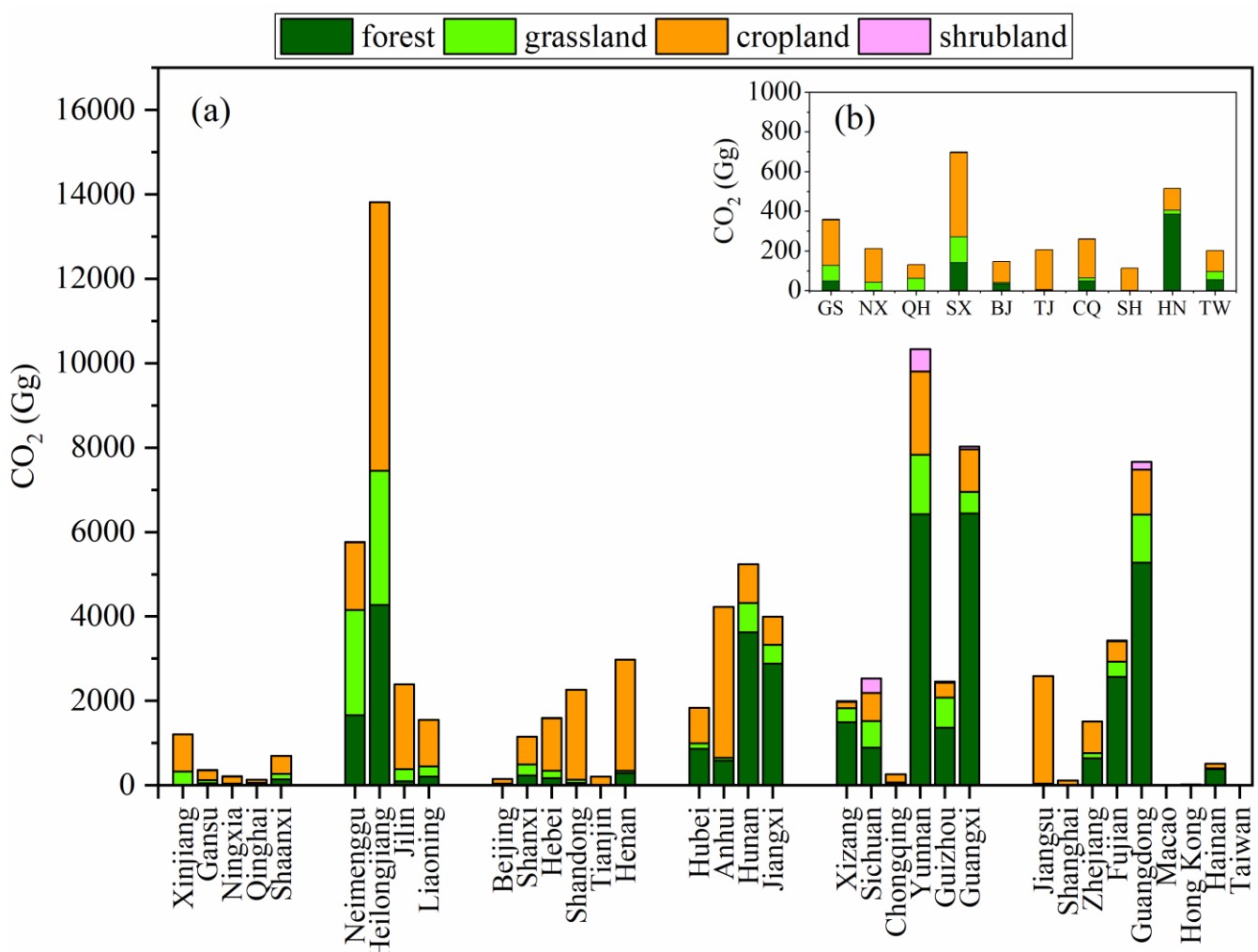

**Figure 1. (a) Source-specific CO₂ emission in each province.** Each group of bars represent a region (from left to right): Northwest (Xinjiang–Shaanxi), Northeast (Neimenggu–Liaoning), North (Beijing–Henan), Central (Hubei–Jiangxi), Southwest (Xizang–Guangxi), Southeast (Jiangsu–Taiwan). **Ten provinces and municipalities with emissions lower than 1000 Gg yr⁻¹ were shown in detail in (b)**: Gansu (GS), Ningxia (NX), Qinghai (QH), Shaanxi (SX), Beijing (BJ), Tianjin (TJ), Chongqing (CQ), Shanghai (SH), Hainan (HN) and Taiwan (TW). Macao and Hong Kong have minimal emissions, that is 0.5 Gg in Macao, consisting of 0.4 Gg from forest fires and 0.1 Gg from grassland fires; and 24.7 Gg in Hong Kong, consisting of 20.6 Gg (83 %) from forest fires, 2.7 Gg (11 %) from grassland fires, 0.7 Gg from shrubland and 0.7 Gg from cropland fires.

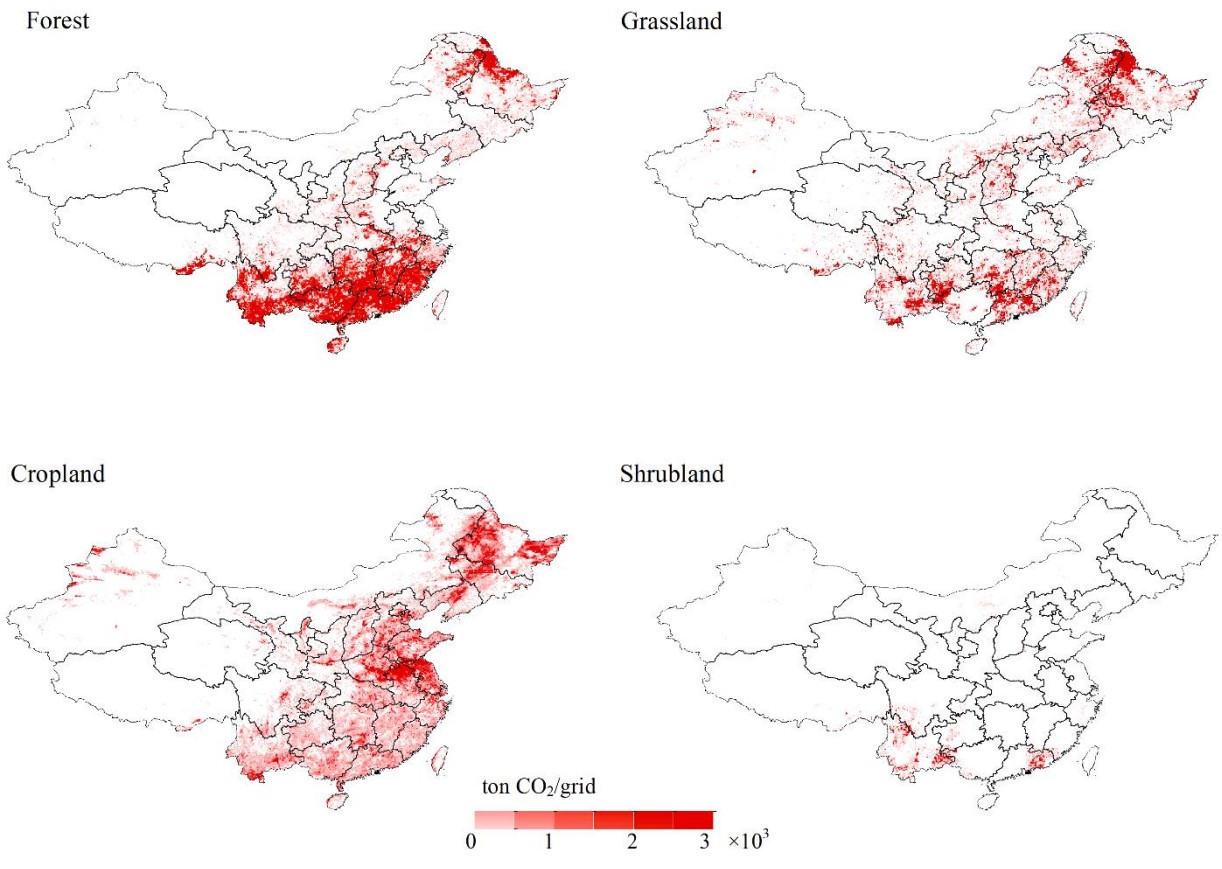

**Figure 2. Spatial distribution of CO₂ emissions (ton) from each land cover type (excluding small islands in the South China Sea).**

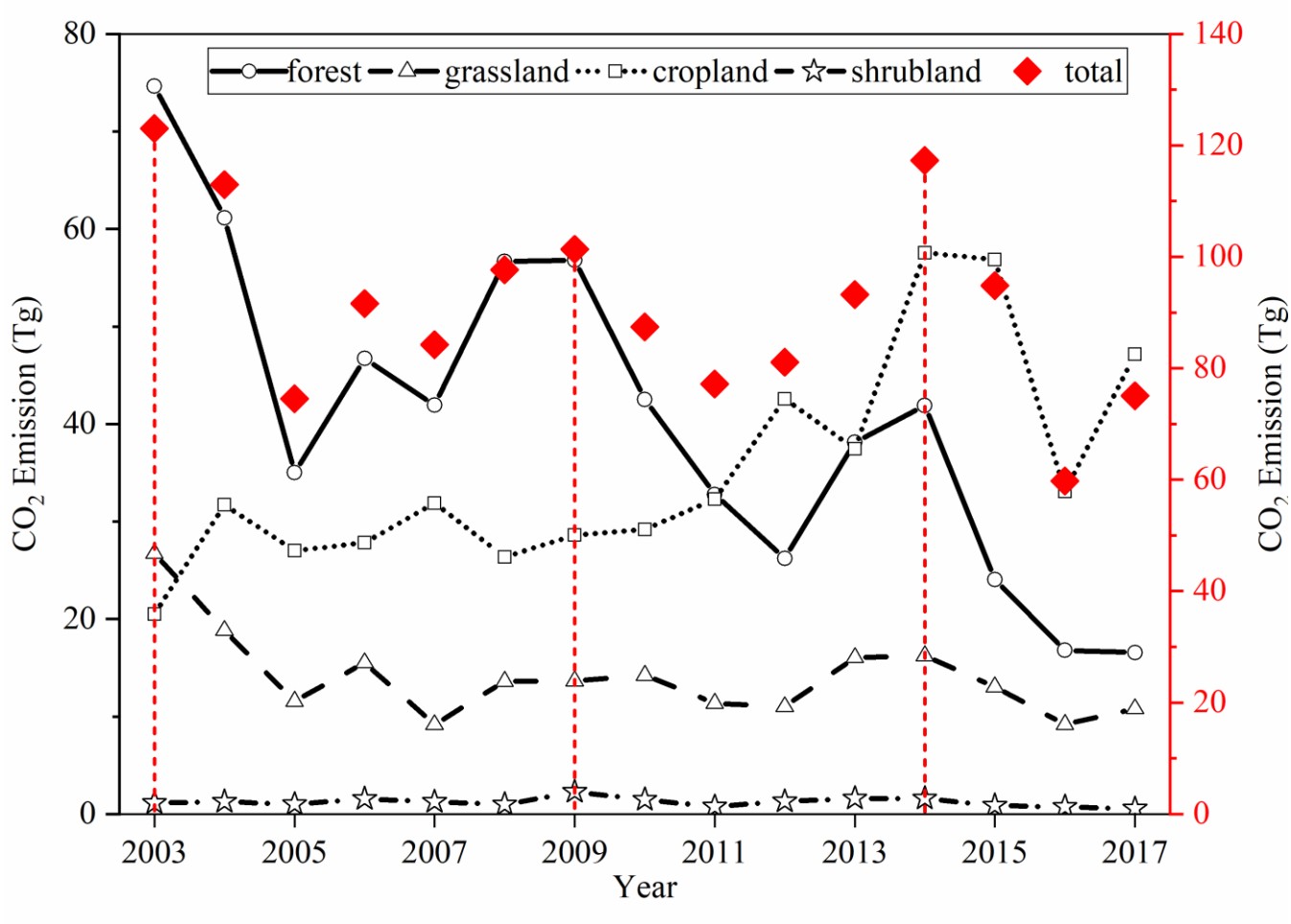

**Figure 3. Annual variation in total and source-specific CO₂ emissions (Tg), 2003–2017**

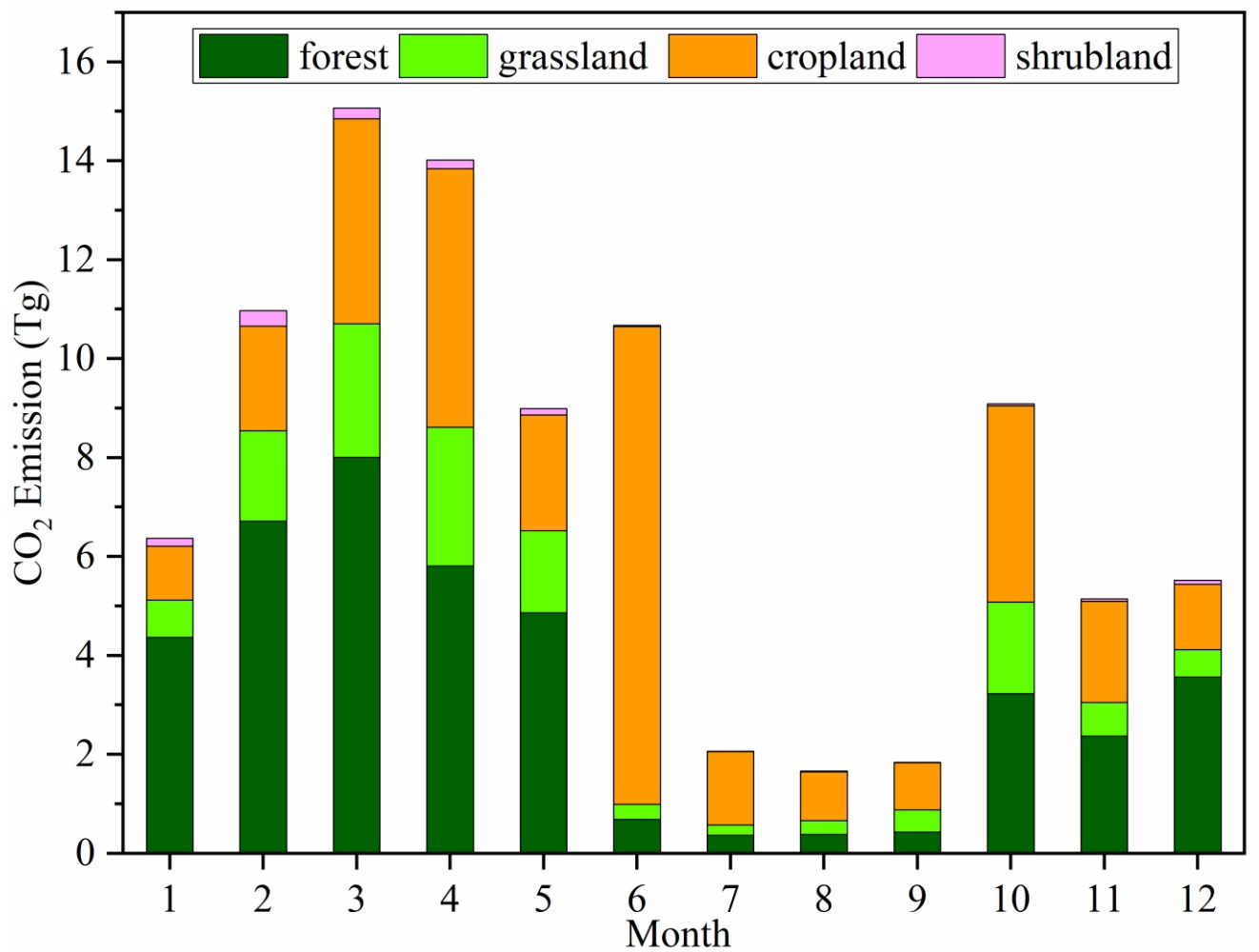

**Figure 4. Monthly distributions of source-specific CO$_2$ emissions (Tg) in China.**