# Peer review of "Estimation of emissions from biomass burning in China (2003–2017) based on MODIS fire radiative energy data"

_Biogeosciences, 2018_

## Referee Comment (RC1) · Anonymous Referee #1 · 11 Jan 2019

In this article, the authors present a study of burning emission in China using MODIS inputs and an empirical fire radiative energy method. The fire emission issue in China is an important one due to its complexity caused by the rapid social development. This paper is well written. But There are still some issues to be discussed.

1. The land cover

Globcover 2009 is used in this study. However, during the study period, China experienced dramatic changes, including urban expansion brought about by rapid urbanization, as well as returning farmland to forests and grasslands. The emission factors are dependent on the type of vegetation. Therefore, at 1km resolution level, large emission

uncertainties may occur due to the biases in land cover data. Maybe annual land cover data is a better choice.

2. Seasonal patterns

The authors did not show much about seasonal patterns of the results, which is very effective in evaluating the results. Due to the impact of the monsoon climate, the meteorological conditions that trigger the fire are extremely seasonal. Meanwhile, the agriculture schedules are very stable in the eastern and northeast plains, and fires from cropland only occur in and after the harvest seasons. For example the results shown in Fig.2, based on personal experience, I am very worried about the confusion of grassland and cropland fires.

3. Monte Carlo

Please explain more details in the Monte Carlo simulations, which (how many) independent variables are fitted and randomly sampled.

4. Double check the words, including $CO_2$ (2 subscript). Use "dry season" rather than "arid season", different meanings.

---

## Referee Comment (RC2) · Anonymous Referee #2 · 16 Jan 2019

Review comment: reject. This study developed a biomass burning emission inventory for China from 2003 to 2017 using a method based on FRE and presented the change of biomass burning emissions from different land cover types. The reason for rejection is that the biomass burning emissions from different land cover types are not reasonable. The aim of this study is to construct an inventory of biomass burning emission for China that could be used in global and regional air quality modeling. So the accuracy of the emission data is very important for data users. In addition, the improving estimation of biomass burning emissions in this study over other estimations was not well proved. The main problem is that the land cover data used in the study is of low accuracy over China. Emission factors for each land-cover type were derived from pub-

lished studies. So the emissions of different gases were related with land cover types. The quality of the land cover data decided the quality of the emission estimation. If the land cover data were not accurate, the results wouldn't be credible. However, all of the discussions were based on the inventory. The spatial distribution of $CO_2$ emissions is not reasonable, especially the distribution from grassland, cropland, and shrublands. If you have read the papers about Chinese land cover written by Zengxiang Zhang et al (2014) and Jun Chen et al., (2016), you could find out that the distribution of the three land cover types were different from what the paper showed. I suggest that you could change the land cover data and improve your writing.

Some detailed comments and suggestions are listed as follows: The introduction didn't have a good logic. The aim of the study is to develop a biomass burning emission inventory for China from 2003 to 2017. The method based on FRE was not innovated by this study, but most of the introduction was about the most often used methods and approach based on FRE. In the introduction, the paper didn't provide a summary about the existing studies for Chinese biomass burning. Maybe some studies were mentioned when the method based on FRE was introduced. Although you mentioned that "few studies have used this approach to estimate emissions from agricultural burning on a national scale", the words didn't support this conclusion. What about other land cover types? What about regional scale? What's more, the cropland distributed intensively in several plains in China. It is not necessary to estimate emissions on a national scale if there are already some studies focusing on the main agricultural regions. Actually, the paper also studied other land cover types. Agricultural burning was not the only study aim. Page 1: line 10: what does "available emission factors" mean specifically? Page 1: line 12: The paper didn't show how the method based on FRE provides a more reasonable estimate from small fires directly. Page 1: line 19-21: this conclusion is not special for this study, so you don't have to put it here. Page 2: line 7: a reference or link need to be added here. I doubted that biomass burning from crop residues leading to substantial pollutant emissions in China. The paper concluded that forest was the major source of biomass burning in China. Page 3: line 12: to prove a method to be

valid should base on field survey, not a comparison with results from another research. Page 3: line 12: it will be better to put "According to the accumulated temperature, China is..." into a new paragraph. Page 3: line 15: is the method that parameterizes the FRP diurnal cycle for crop zones and harvest seasons innovated by you, or it was proposed by former studies? If it was proposed by you, you should put it in the method section. If it was proposed by former study, a reference should be provided. Page 3: line 23: introduction about the global land cover data should be put in the data section. Page 4: line 10-12: why did you use the average value, not one of themïïj§When the two values were provided, didn't the researchers give suggestions about their applications? As the CR was very important in calculating the emissions, the value should be decided more carefully. Page 4: line 16: the method section should introduce the method used in the study and how you used the method to get the results, not the method provided by the former research. The expression should be improved. Page 4: line 24: that "the origin formula couldn't provide reasonable estimations" and that "h has little effect on the final calculation" seems to be contradictory. Page 5: line 2: is $\varepsilon$ a constant or variable? Maybe a variable, as you didn't present its value. If it was a variable, how did you decide its value? Page 5: line 14: the expression is not accurate. GlobCover maybe the most detailed map of earth land surface at the same spatial resolution. The reference was not the newest. Many new land cover datasets have been produced in recent decades. Maybe other land cover datasets like Globeland30 (Jun Chen et al., 2016) or NLUD-C (Zengxiang Zhang et al., 2014) are more suitable. The four main land cover types used in this study could be found in this dataset. And the accuracy of Globeland30 is better than GLobCover 2009 since it has higher spatial resolution. Page 7: line 7: if figure 2 was presented on a national province map, it would be clearer that how the emissions distribute in different provinces. A land cover map can be presented simultaneously. Page 9: line 16: to decide if the results are reasonable or not, you should compare the calculating results with field data or the statistical data from government, not just compare it with other research data. Page 9: line 18: if you mean that the discrepancy between the former studies (GFED4s and

GFASv1) and your results is caused by the high omission rate of small fires in the two existing datasets, then you should prove this by comparing the two results directly, not just by citing a reference. Page 10: line 10: although the paper concluded that the estimation of biomass burning emissions in this study was improved, it was hard to confirm its credibility. As the words "perhaps due to" were used in this paper. Page 10: line 20: in this paragraph, you mentioned several sources of errors. The Monte Carlo simulations seemed to calculate the uncertainty caused by emission factors. What about the uncertainties caused by other error sources? Page 11: line 13: if your estimates were just very close to the results from GFED4s and GFASv1.0, then why the users would choose your estimations?

---

## Author Comment (AC1) · 28 Feb 2019

**Response to referee comment #1:**

*In this article, the authors present a study of burning emission in China using MODIS inputs and an empirical fire radiative energy method. The fire emission issue in China is an important one due to its complexity caused by the rapid social development. This paper is well written. But There are still some issues to be discussed.*

**Response:** We appreciate the review's comments, which indeed help us to improve the manuscript much.

*1. The land cover*
*Globcover 2009 is used in this study. However, during the study period, China experienced dramatic changes, including urban expansion brought about by rapid urbanization, as well as returning farmland to forests and grasslands. The emission factors are dependent on the type of vegetation. Therefore, at 1km resolution level, large emission uncertainties may occur due to the biases in land cover data. Maybe annual land cover data is a better choice.*

**Response:** Accepted. MODIS land cover product (MCD12Q1) provide annual land cover data, but the spatial resolution (500 m) is relatively coarse. The open fire size is often small in China, we used GlobelLand30 dataset with 30 m resolution (covering year 2000 and 2010) instead of GlobCover2009 in the revised manuscript. The land cover types are characterized by GlobeLand30-2000 for years 2003-2005 and GlobeLand30-2010 for years 2006-2017.

**Revisions:** (Page 6, Line 9) "The GlobeLand30 dataset maps global land cover at 30 m spatial resolution in two base years (2000 and 2010) (Chen et al., 2017b), as shown in Fig S1. GlobeLand30 data are generated by multispectral images derived from Landsat TM, ETM+ and Chinese Environmental Disaster Alleviation Satellite (HJ-1). The result of accuracy assessment shows that the overall accuracy of GlobeLand30 reaches 83.5 %. GlobeLand30 dataset consists of 10 land cover types, namely cultivated land, forest, grassland, shrubland, wetland, water bodies, tundra, artificial surfaces, bareland, permanent snow and ice. In this study, the land cover types are characterized by GlobeLand30-2000 for years 2003-2005 and GlobeLand30-2010 for years 2006-2017. We combined the land-cover map of China and the latitude and longitude data of fire count in MOD14/MYD14 to determine the biomass fuel types. For instance, if a fire count locates in cropland area, it will be considered as a crop residue burning event."

*2. Seasonal patterns*
*The authors did not show much about seasonal patterns of the results, which is very effective in evaluating the results. Due to the impact of the monsoon climate, the meteorological conditions that trigger the fire are extremely seasonal. Meanwhile, the agriculture schedules are very stable in the eastern and northeast plains, and fires from cropland only occur in and after the harvest seasons. For example the results shown in*

*Fig.2, based on personal experience, I am very worried about the confusion of grassland and cropland fires.*

**Response:** Accepted. Seasonal patterns were introduced in Section 3.2 and Fig.4. The spatial distributions of grassland and cropland fires shown in Fig.2 are reasonable. Grassland fires are mainly distributed in the mountains and hills in northeastern China and southern China. Cropland fires are concentrated in central China and northeastern China due the burning of winter wheat residue in the North China Plain and corn straw in the Northeast China Plain.

**Revisions:** (Page 9, Line 17) "Seasonal variations of $CO_2$ emissions from each source were presented in Fig.4. In terms of total emissions, spring (March, April and May) contributed the most emissions due to the impact of dry weather. The lowest emissions occurred in rainy season including July, August, and September, producing 2.1,1.7, and 1.8 Tg $CO_2$, respectively. From the perspective of source-specific emissions, forest and grassland fires exhibited similar temporal variation, i.e., higher emissions in winter and spring, and lower emissions in summer. The highest emissions from forest and grassland fires occurred in the period of January to May. This pattern was strongly affected by favorable fire conditions such as low vegetation moisture content and high wind speed (Song et al., 2009). In addition, Li et al. (2015) found that a large portion of forest fires in spring were induced by sacrificial activity in Tomb-sweeping Day (April 5). Forest Fires in winter were concentrated in southern China due to the impacts of low precipitation and mild temperatures. In contrast, boreal forests rarely burned because of the low temperatures and moist snow cover. This result was consistent with the that reported by Chen et al. (2017a). The temporal distribution of shrubland fire emissions is also similar to that of forest and grassland fires, but emissions from bush only account for a small fraction of total levels (approximately 1 %). Emissions from crop burning were closely related to agriculture activities. Different main crops and sowing/harvest times in different areas lead to multiple emission peaks (Jin et al., 2018). Highest emissions occurred in summer, and small peaks were detected in spring and autumn. Emissions from agriculture fires contribute 84 % to total emissions in summer, which were concentrated in June due to the large amount of winter wheat straw burning in the North China Plain. From March to May, as large amounts of crop residues were burned to clear the cultivated land for sowing, fires were scattered throughout the country. In autumn (especially October), corn straws burning in the Northeast China Plain and late rice residue burning in southern China were primary contributors, and small areas of maize residue burning could be found in northern China (Chen et al., 2017a). During winter, crop burning

mostly occurred in southern China due to citrus harvest and orchard clearing activity."

*3. Monte Carlo*

*Please explain more details in the Monte Carlo simulations, which (how many) independent variables are fitted and randomly sampled.*

**Response:** Accepted. More details could be found in Section 4.

**Revisions:** (Page 12, Line 6) "In this study, we considered errors of three independent variables, namely FRE, conversion ratio and emission factors. According to the error budget suggested by Vermote et al. (2009), we assumed that the relative error of FRE and the conversion ratio was 31 % and 10 %, respectively. The uncertainty of the EF is species dependent and we applied the uncertainty suggested in Huang et al. (2012), as shown in Table S2. We ran 20,000 Monte Carlo simulations to estimate the range of average annual fire emissions in 2003-2017 with a 90 % confidence interval. In Monte Carlo simulation, random number are selected from normal distribution of input variables."

*4. Double check the words, including CO2 (2 subscript). Use "dry season" rather than "arid season", different meanings.*

**Response:** Accepted.

**Revisions:** (Page 1, Line 20) "Forest and grassland fires are concentrated in northeast and south China, especially in dry season (from October to March of the following year)."

---

## Author Comment (AC2) · 28 Feb 2019

**Response to referee comment #2:**

*Review comment: reject. This study developed a biomass burning emission inventory for China from 2003 to 2017 using a method based on FRE and presented the change of biomass burning emissions from different land cover types. The reason for rejection is that the biomass burning emissions from different land cover types are not reasonable. The aim of this study is to construct an inventory of biomass burning emission for China that could be used in global and regional air quality modeling. So the accuracy of the emission data is very important for data users. In addition, the improving estimation of biomass burning emissions in this study over other estimations was not well proved. The main problem is that the land cover data used in the study is of low accuracy over China. Emission factors for each land-cover type were derived from published studies. So the emissions of different gases were related with land cover types. The quality of the land cover data decided the quality of the emission estimation. If the land cover data were not accurate, the results wouldn't be credible. However, all of the discussions were based on the inventory. The spatial distribution of CO2 emissions is not reasonable, especially the distribution from grassland, cropland, and shrublands. If you have read the papers about Chinese land cover written by Zengxiang Zhang et al (2014) and Jun Chen et al., (2016), you could find out that the distribution of the three land cover types were different from what the paper showed. I suggest that you could change the land cover data and improve your writing.*

**Response:** We appreciate the review's comments, which indeed help us to improve the manuscript much. As the reviewer suggested, we used GlobeLand30 datasets instead of GlobCover 2009 in revised manuscript. GlobeLand30 maps global land cover at 30 m spatial resolution in two base years (2000 and 2010). In this study, the land cover types are characterized by GlobeLand30-2000 for years 2003-2005 and GlobeLand30-2010 for years 2006-2017. We would let you know that the NLUD-C (Zengxiang Zhang et al., 2014) is commercial at present, and the price is more than 80 thousand U.S. dollars for three years. We cannot afford them.

*Some detailed comments and suggestions are listed as follows: The introduction didn't have a good logic. The aim of the study is to develop a biomass burning emission inventory for China from 2003 to 2017. The method based on FRE was not innovated by this study, but most of the introduction was about the most often used methods and approach based on FRE. In the introduction, the paper didn't provide a summary about the existing studies for Chinese biomass burning. Maybe some studies were mentioned when the method based on FRE was introduced. Although you mentioned that "few studies have used this approach to estimate emissions from agricultural burning on a national scale", the words didn't support this conclusion. What about other land cover types? What about regional scale? What's more, the cropland distributed intensively in several plains in China. It is not necessary to estimate emissions on a national scale if there are already some studies focusing on the main agricultural regions. Actually, the paper also studied other land cover types. Agricultural burning was not the only study*

*aim.*

**Response:** Accepted. We reworded the Introduction section according to your comments.

[revised manuscript text omitted]

*Page 1: line 10: what does "available emission factors" mean specifically?*

**Response:** The emission factors we used in this study were cited from references listed in Supplement (Table S1).

**Revisions:** (Supplement, Table S1) "Note: Superscript letters indicate the data source. Sources are from the following: [a] (Andreae and Merlet, 2001; Streets et al., 2003; Cao et al., 2004; Michel et al., 2005; Wiedinmyer et al., 2006). [b] (Andreae and Merlet, 2001; Streets et al., 2001; Reddy and Venkataraman, 2002; Cao et al., 2006)."

*Page 1: line 12: The paper didn't show how the method based on FRE provides a more reasonable estimate from small fires directly.*

**Response:** Accepted. Detailed information had been added in Introduction section.

**Revisions:** (Page 2, Line 23) "… as the average cultivated area of a farming household is very limited in China (around $10^4$ m$^2$), each agricultural fire burns within a small extent (Liu et al., 2015). Therefore, the fire count method is likely to overestimate the burned area of crop residue burning, and these fires are not detected efficiently by the available burned area algorithms due to the small areas and intermittency (Song et al., 2009). For a better estimation of biomass burning emission, an approach based on fire radiative energy (FRE) was proposed as a new tool for global studies of vegetation fires around the year 2000 (Kaufman et al., 1996; Wooster, 2002). The FRE method estimates

biomass burned amount according to energy radiated from fires, which could avoid the uncertainty caused by inaccuracy of satellite-derived burned area and therefore improve the estimation, especially for small fire emissions."

(Page 3, Line 9) "… the amount of pollutants released by biomass burning could be calculated as a product of FRE, conversion ratio and emission factors, reducing uncertainties from multiple parameters that are not reliably defined at regional and global scales (Wooster et al., 2005)."

*Page 1: line 19-21: this conclusion is not special for this study, so you don't have to put it here.*

**Response:** Accepted. We improved the expression of our conclusion.

**Revisions:** (Page 1, Line 20) "Forest and grassland fires are concentrated in northeastern China and southern China, especially in dry season (from October to March of the following year). Plain areas with high crop yields, such as the North China Plain, experienced high agricultural fire emissions in harvest seasons. Most shrubland fires located in Yunnan and Guangdong province."

*Page 2: line 7: a reference or link need to be added here. I doubted that biomass burning from crop residues leading to substantial pollutant emissions in China. The paper concluded that forest was the major source of biomass burning in China.*

**Response:** Accepted. We added a reference here (Streets et al., 2003). In the revised manuscript, GlobeLand30 was used to characterize biomass type and the results still showed that forest fires contribute most to the total emissions. Actually, annual mean emissions from forest fires and crop residue burning are very closed (40.8 and 35.3 Tg $CO_2$, respectively). Most studies tend to focus on the emissions in eastern China and central China or densely populated regions to investigate the impacts of pollutants on human health. Crop residue burning extensively occurred in these areas due to the developed agriculture. In our study, agricultural fires are also determined to be the primary contributor in northeastern and central regions, accounting for 55 % of the total $CO_2$ emissions. However, forest fires are concentrated in north and southwest provinces, which are remote and sparsely populated. According to Yan et al. (2006) , forest fires are substantially understated in official statistical data. Due to the high biomass density, forest fires would release large quantities of pollutants. Studies based on satellite burned area products reported that annual $CO_2$ emissions range from 19 to 137 Tg (Qiu et al., 2016;Song et al., 2009), which could account for a large portion of total biomass burning emission. Emissions caused by forest fires has not been studied in great detail.

**Revisions:** (Page 2, Line 7) "The annual amount of crop residue burned in fields estimated by Streets et al. (2003) was 110 Tg, accounting for 44 % of all crop residue burned in Asia, leading to substantial pollutant emissions."

(Page 7, Line 12) "On a national scale, forest fires contribute the largest portion (45 %) of total $CO_2$ emissions from open fires. Agricultural fires and grassland fires ranked for the second and third places, accounting for 39 % and 15 %, respectively. Regionally, the main emission contributor is different. In southwestern region, the percentage of emission from forest fires could reach up to about 65 %, whereas the most important source in northeastern China is crop residue burning, accounting for 47 % of total emissions. The result was in connection with rural population intensity and land use patterns (Qiu et al., 2016). For example, due to the dense boreal forests and developed agriculture, the highest emission was found in Heilongjiang with 46 % from agriculture fires and 54 % from forest and grassland fires. Similarly, in the southwestern region, the dense vegetative cover of Yunnan-Guizhou Plateau greatly contributes to fire events. In northern and central regions, approximately 55 % of fire emissions were derived from agricultural fires. Benefiting from fertile land and favorable climate, the two regions contain many principal agricultural provinces (including Shandong, Henan, Hubei and Anhui Provinces) and therefore large amounts of crop residue were burned in field during the harvest season."

*Page 3: line 12: to prove a method to be valid should base on field survey, not a comparison with results from another research.*

**Response:** Accepted. Results of Huang et al. (2012) were based on statistical data, and we reworded the sentence.

**Revisions:** (Page 3, Line 11) "Liu et al. (2015) applied FRE approach to estimate emissions of crop residue burning in North China Plain during the harvest season (June). The differences of their results with those derived from official statistical data (Huang et al., 2012) were mostly around -13 % with the largest difference of -49 %. Besides, their results were significantly higher than those derived from burned area method. These comparisons suggested that the approach produced a reasonable estimation."

*Page 3: line 12: it will be better to put "According to the accumulated temperature, China is. . ." into a new paragraph.*

**Response:** Accepted.

*Page 3: line 15: is the method that parameterizes the FRP diurnal cycle for crop zones and harvest seasons innovated by you, or it was proposed by former studies? If it was proposed by you, you should put it in the method section. If it was proposed by former*

*study, a reference should be provided.*

**Response:** Accepted. Detailed information had been added in Section 2.1.

**Revisions:** (Page 5, Line 17) "Different combustion characteristics of fuel types could be reflected by specific T/A ratio. Because the dominate crop type vary greatly among temperature zones, we divide the country into six regions (tropical zone, subtropical zone, warm-/middle-/cold–temperate zone and Qinghai-Tibet plateau) and calculate the monthly T/A ratio separately. Using respective T/A ratio to calculate factors in Eq. (2), the FRP diurnal cycle was parameterized for each crop zone and harvest season, which could reflect specific combustion characteristic for different straw types."

*Page 3: line 23: introduction about the global land cover data should be put in the data section.*

**Response:** Accepted.

**Revisions:** (Page 6, Line 9) "The GlobeLand30 dataset maps global land cover at 30 m spatial resolution in two base years (2000 and 2010) (Chen et al., 2017b), as shown in Fig S1. GlobeLand30 data are generated by multispectral images derived from Landsat TM, ETM+ and Chinese Environmental Disaster Alleviation Satellite (HJ-1). The result of accuracy assessment shows that the overall accuracy of GlobeLand30 reaches 83.5 %. GlobeLand30 dataset consists of 10 land cover types, namely cultivated land, forest, grassland, shrubland, wetland, water bodies, tundra, artificial surfaces, bareland, permanent snow and ice. In this study, the land cover types are characterized by GlobeLand30-2000 for years 2003-2005 and GlobeLand30-2010 for years 2006-2017. We combined the land-cover map of China and the latitude and longitude data of fire count in MOD14/MYD14 to determine the biomass fuel types. For instance, if a fire count locates in cropland area, it will be considered as a crop residue burning event."

*Page 4: line 10-12: why did you use the average value, not one of them? When the two values were provided, didn't the researchers give suggestions about their applications? As the CR was very important in calculating the emissions, the value should be decided more carefully.*

**Response:** The field experiment conducted by Wooster et al. (2005) tried to replicate conditions of dry season savanna fires. The fuels used were Miscanthus, dried grasses, wheat stems and a woody fuel. The experiment of Freeborn et al. (2008) was conducted in combustion chamber, and most of the fuels used are wood and herbaceous. When Vermote et al. (2009) developed the modified Gaussian function to calculate FRE and estimate global biomass burning emissions, they applied the average of conversion factors from above two studies. We adopted the mean value for calculation referred to method of

Vermote et al. (2009). The potential error of conversion factor has been considered in Monte Carlo simulations.

**Revisions:** (Page 12, Line 7) "According to the error budget suggested by Vermote et al. (2009), we assumed that the relative error of FRE and the conversion ratio was 31 % and 10 %, respectively."

*Page 4: line 16: the method section should introduce the method used in the study and how you used the method to get the results, not the method provided by the former research. The expression should be improved.*

**Response:** Accepted. We improved the expression as you suggested.

**Revisions:** (Page 4, Line 8) "Pollutant emissions were calculated as the product of dry mass burned (kg) and a corresponding emission factor (g kg$^{-1}$). In this study, emission factors for each land cover type were obtained from previous publications (Table S1). If more than one value for an emission factor is available, the average value is used. The amount of biomass consumed was calculated by multiplying FRE by a conversion ratio, which was not significantly influenced by vegetation types"

(Page 4, Line 18) "FRE was estimated by integrating FRP (i.e. instantaneous FRE) over the duration of the fire process. In this study, FRP data from MODIS active fire products (MOD14/MYD14) were used. The MODIS sensors, onboard the polar-orbiting satellites Terra and Aqua, acquire four discrete FRP data at 1030/2230 (Terra) and 0130/1330 (Aqua), equatorial local time. Therefore, the fire diurnal variation cannot be directly detected by satellite observation and many fire events have been missed. To calculate FRE and make up the omission error, we used a modified Gaussian function (Vermote et al., 2009) to parameterize the FRP diurnal cycle as Eq. (2). This parameterization describes the discrete observations as a continuous function and simplifies integral process to calculate total fire energy released."

*Page 4: line 24: that "the origin formula couldn't provide reasonable estimations" and that "h has little effect on the final calculation" seems to be contradictory.*

**Response:** Accepted. We reworded the sentences.

**Revisions:** (Page 5, Line 12) "We found that the original parameterized FRP diurnal cycle could not agree well with the observed FRP temporal variation in China, possibly due to inaccurate FRP peak hour. Because it has been pointed that *h* has little effect on the final result of FRE (Vermote et al., 2009), we added a parameter $\varepsilon$ ($\varepsilon$=4) in order to modify FRP peak hour (Liu et al., 2015)."

*Page 5: line 2: is $\varepsilon$ a constant or variable? Maybe a variable, as you didn't present its value. If it was a variable, how did you decide its value?*

**Response:** Accepted.

**Revisions:** (Page 5, Line 13) "Because it has been pointed that *h* has little effect on the final result of FRE (Vermote et al., 2009), we added a parameter $\varepsilon$ ($\varepsilon$=4) in order to modify FRP peak hour (Liu et al., 2015)."

*Page 5: line 14: the expression is not accurate. GlobCover maybe the most detailed map of earth land surface at the same spatial resolution. The reference was not the newest. Many new land cover datasets have been produced in recent decades. Maybe other land cover datasets like Globeland30 (Jun Chen et al., 2016) or NLUD-C (Zengxiang Zhang et al., 2014) are more suitable. The four main land cover types used in this study could be found in this dataset. And the accuracy of Globeland30 is better than GLobCover 2009 since it has higher spatial resolution.*

**Response:** Accepted. As you suggested, we used GlobeLand30 datasets instead of GlobCover 2009 in revised manuscript. GlobeLand30 maps global land cover at 30 m spatial resolution in two base years (2000 and 2010). In this study, the land cover types are characterized by GlobeLand30-2000 for years 2003-2005 and GlobeLand30-2010 for years 2006-2017.

*Page 7: line 7: if figure 2 was presented on a national province map, it would be clearer that how the emissions distribute in different provinces. A land cover map can be presented simultaneously.*

**Response:** Accepted. We presented Figure 2 on provincial level. The land cover map was presented in Figure S1. It is difficult to present them in one figure simultaneously as land cover map would be overlaid by emission distribution map.

*20. Page 9: line 16: to decide if the results are reasonable or not, you should compare the calculating results with field data or the statistical data from government, not just compare it with other research data.*

**Response:** Accepted. Comparisons of our results with those obtained from statistical data were shown in Section 3.2 and Section 3.3.

**Revisions:** (Page 8, Line 17) "Peak emissions occurred in 2003, 2009 and 2014; forest fires in 2003 and 2009, and cropland fires in 2014 were determined to be the primary contributors, accounting for 61 %, 56 %, and 49 % of total emissions in that year, respectively. Our results were in accordance with the records reported in official statistics. According to the China Forestry Statistical Yearbook, there are seven extraordinarily serious fire accidents in 2003, resulting in the largest forest burned area during the study period. A total of 35 serious fire accidents happened in 2009, 171 % higher than the 15-year average number of that kind of fire events (12.9)."

(Page 9, Line 1) "Pollutants released by crop straw burning continue to rise in 2003-2014, leading to a peak emission of 57.6 Tg $CO_2$ in 2014. Because crop residues burning in field could be well controlled by strict supervision,

cropland emissions have decreased rapidly in 2015-2016 (dropped by 42 %). However, the emissions increased again by 37 % in 2017. This variation trend was similar to that concluded by studies based on statistical data (Li et al., 2016b; Jian et al., 2018). Yan et al. (2006) pointed that as the socioeconomic development, which results in a decline of biofuel (crop residue, fuel wood) demand, crop residue is increasingly being burned in the field. Tao et al. (2018) found that the consumption of crop residues as residential energy in rural China decreased by 51 % from 1992 to 2012. We note that the number of agricultural fire count increased by a factor of 3 in 2003-2014 (from 13683 to 67143), which could support the conclusion as well. Although the controlling of pollutants from crop residue burning in China started from 1965, it seems to be ineffective and the crop straw burning should be further focused."

(Page 10, Line 20) "When compared with results of Huang et al. (2012) and Yan et al. (2006), which were based on official statistical data, our results are larger for forest and grassland fires, and underestimated for crop residue burning. According to Yan et al. (2006), forest and grassland fires were understated in statistics for both personal and political reasons. They suggested that satellite data are preferable to statistical data to estimate emissions from forest and grassland fires. When statistics were used to estimate crop residue emission, the crop residue burnt are calculated as a product of crop production, residue-to-production ratio, dry matter-to-crop residue ratio, the percentage of dry matter burned in fields, and combustion efficiency. Values of these parameters depend on local agricultural practices and vary greatly in different studies. For example, the value of percentage of residue burned in field, which is one of the most important factors to be determined, ranges from 6.6 % to 82 % in different research (Gao et al., 2002; Yang et al., 2008; Yan et al., 2006). The accumulation of uncertainties derived from multiple factors could result in significant emission uncertainties. Using statistical data, amount of burned residue was estimated to be 40-160 Tg yr$^{-1}$, showing a great potential error (Li et al., 2016a; Huang et al., 2012). Therefore, results derived from statistical are not necessarily reliable."

*Page 9: line 18: if you mean that the discrepancy between the former studies (GFED4s and GFASv1) and your results is caused by the high omission rate of small fires in the two existing datasets, then you should prove this by comparing the two results directly, not just by citing a reference.*

**Response:** Our results were closed to those derived from GFED4s and GFASv1, but substantially higher than those based on the burned area product (MCD64A1). As the FRE method calculate emissions based on radiated energy from fires,

burned area data cannot be obtained from FRE method. It is difficult to compare the burned area data from two methods directly. However, when used data in MCD64A1 to make estimate, the burned area is the most important factor in determining fire emissions. Therefore, the underestimation in results from MCD64A1 can primarily attribute to the high omission rate of small fires and uncertainty in the calculation of fire-affected area.

**Revisions:** (Page 10, Line 15) "… as shown in Table 3, results calculated by using data from burned area product MCD64A1 were substantially underestimated. In this method, burned area is one of the most important factors in calculating emissions, so that the underestimation could be attributed to omission of fires with small areas and short duration (Song et al., 2009)."

*Page 10: line 10: although the paper concluded that the estimation of biomass burning emissions in this study was improved, it was hard to confirm its credibility. As the words "perhaps due to" were used in this paper.*

**Response:** Accepted. We improved the expression. We concluded that our estimation was improved by the comparison of results based on FRE method with those based on statistical and satellite data.

**Revisions:** (Page 10, Line 20) "When compared with results of Huang et al. (2012) and Yan et al. (2006), which were based on official statistical data, our results are larger for forest and grassland fires, and underestimated for crop residue burning. According to Yan et al. (2006), forest and grassland fires were understated in statistics for both personal and political reasons. They suggested that satellite data are preferable to statistical data to estimate emissions from forest and grassland fires. When statistics were used to estimate crop residue emission, the crop residue burnt are calculated as a product of crop production, residue-to-production ratio, dry matter-to-crop residue ratio, the percentage of dry matter burned in fields, and combustion efficiency. Values of these parameters depend on local agricultural practices and vary greatly in different studies. For example, the value of percentage of residue burned in field, which is one of the most important factors to be determined, ranges from 6.6 % to 82 % in different research (Gao et al., 2002; Yang et al., 2008; Yan et al., 2006). The accumulation of uncertainties derived from multiple factors could result in significant emission uncertainties. Using statistical data, amount of burned residue was estimated to be 40-160 Tg $y^{r-1}$, showing a great potential error (Li et al., 2016a; Huang et al., 2012). Therefore, results derived from statistical are not necessarily reliable. When compared to other inventories based on remote sensing data, our results agree well with those reported by GFED4s and substantially higher than those derived from burned area product (MCD64A1). Datasets in GFED4s are based on burned area boosted by small fire burned area, which could provide a relatively high emission estimation of agricultural fires."

(Page 11, Line 18) "Our results were higher than those based on burned area products as the FRE method avoids uncertainties cause by inaccuracy of satellite-derived burned area and multiple other parameters. The results were closed to those derived from FINNv1.5 in terms of emissions from grassland and cropland fires and accorded with those from GFED4s for all fire types. The temporal and spatial resolution of our inventory (daily, 1 km) are higher than that of GFED4s (monthly, 0.25 degrees) and GFASv1.0 (daily, 0.5 degrees). Compared with other inventories, we considered specific combustion characteristics of different crop types and calculated the agriculture fires emissions separately according to the distribution of temperate zones. Therefore, this method developed a high-resolution inventory and improved estimation of biomass burning emissions, especially for small fires in cropland."

*Page 10: line 20: in this paragraph, you mentioned several sources of errors. The Monte Carlo simulations seemed to calculate the uncertainty caused by emission factors. What about the uncertainties caused by other error sources?*

**Response:** Uncertainties of three independent variables are considered in Monte Carlo simulations, namely FRE, conversion ratio and emission factors. More details could be found in Section 4.

**Revisions:** (Page 12, Line 6) "In this study, we considered errors of three independent variables, including FRE, conversion ratio and emission factors. According to the error budget suggested by Vermote et al. (2009), we assumed that the relative error of FRE and the conversion ratio was 31 % and 10 %, respectively. The uncertainty of the EF is species dependent and we applied the uncertainty suggested in Huang et al. (2012), as shown in Table S2. We ran 20,000 Monte Carlo simulations to estimate the range of average annual fire emissions in 2003-2017 with a 90 % confidence interval. In Monte Carlo simulation, random number are selected from normal distribution of input variables."

*Page 11: line 13: if your estimates were just very close to the results from GFED4s and GFASv1.0, then why the users would choose your estimations?*

**Response:** Considering different combustion characteristics of different crop types, the agriculture fires emissions are calculated separately according to the distribution of temperate zones in our method. Besides, the temporal and spatial resolution of our inventory (daily, 1 km) are higher than that of GFED4s (monthly, 0.25 degrees) and GFASv1.0 (daily, 0.5 degrees).

**Revisions:** (Page 11, Line 19) "The results were closed to those derived from FINNv1.5 in terms of emissions from grassland and cropland fires and accorded with those from GFED4s for all fire types. The temporal and spatial resolution of our inventory (daily, 1 km) are higher than that of GFED4s (monthly, 0.25 degrees) and GFASv1.0 (daily, 0.5 degrees). Compared with other inventories,

we considered specific combustion characteristics of different crop types and calculated the agriculture fires emissions separately according to the distribution of temperate zones. Therefore, this method developed a high-resolution inventory and improved estimation of biomass burning emissions, especially for small fires in cropland."